# A/B TESTING UNDER IDENTITY FRAGMENTATION

## ABSTRACT

Randomized online experimentation is a key cornerstone of the online world. The infrastructure enabling such methodologies is critically dependent on user identification. However, nowadays consumers routinely interact with online businesses across multiple devices which are often recorded with different identifiers for the same consumer. The inability to match different device identities across consumers leads to an incorrect estimation of various causal effects. Moreover, without strong assumptions about the device-user graph, the causal effects are not identifiable. In this paper, we consider the task of estimating global treatment effects (GATE) from a fragmented view of exposures and outcomes. Our experiments validate our theoretical analysis, and estimators obtained through our procedure are shown be superior to standard estimators, with a lower bias and increased robustness.

## 1 INTRODUCTION

A/B testing has become indispensable to online businesses for improving user experience and driving up revenue. The infrastructure which enables this is critically dependent on identifiers, such as cookies or mobile device IDs, traditionally used by websites and apps to track users' browsing behavior and provide personalized content and ads. However, the assumption about the availability of identifiers has become more and more tenuous as users increasingly rely on multiple devices. This means that a customer's effective persona as seen by the advertiser is broken into multiple units – a phenomenon known as 'identity fragmentation'(Coey & Bailey, 2016; Lin & Misra, 2021). Further, the use of third-party identifiers is increasingly being curbed, due to privacy concerns, by both governmental and non-governmental entities, through legislation such as the GDPR [1] and through the deprecation of third-party cookies and advertising identifiers such as the Android Advertising ID (AAID) and the Identifier for Advertisers (IDFA).

Lack of *identifiable information across devices creates a fundamental issue in A/B testing, as the users' exposure to treatment is not fully known in this setting*. Consider the case of a business exploring whether a certain advertisement produces a higher click-through rate. Under the standard A/B testing protocol, a random subset of users will be shown the new ad (B), and the outcome recorded. By comparing the outcomes for these users against the set of users who received ad A, one can estimate the relative change caused in the click-through rate by ad B. For a user who visits using different devices, for instance a smartphone and a tablet, the unique identifier (say IDFA), allows the server to consistently show the user only ad B. However, without identifiers, one cannot be certain of whether the current device should be in the treatment group or the control group. This happens because, while the treatment is administered at device level, the outcomes are dependent on user-level treatments. Thus, the outcome as observed for a device can potentially be affected by the treatment on other devices. This constitutes a *violation of the stable unit treatment – SUTVA assumption (Rubin, 1980) – which standard A/B testing relies upon*.

This phenomenon of treatments to a unit affecting outcomes for other units has been studied in causal literature (Hudgens & Halloran, 2008; LeSage & Pace, 2009) under the name of interference. It is also known as spillover, due to treatment exposure 'spilling over' from one unit to another. However, most methods involving spillover, assume strong restrictions on the structure of spillover (Ogburn et al., 2017; Leung, 2020). The deprecation of identifiers *introduces a new scenario, requiring the estimation of treatment effects from an uncertain interference structure*. This problem setting involves new assumptions compared to prior work. Notably, in addition to the assumption that unit/device

---

[1] https://gdpr-info.eu/

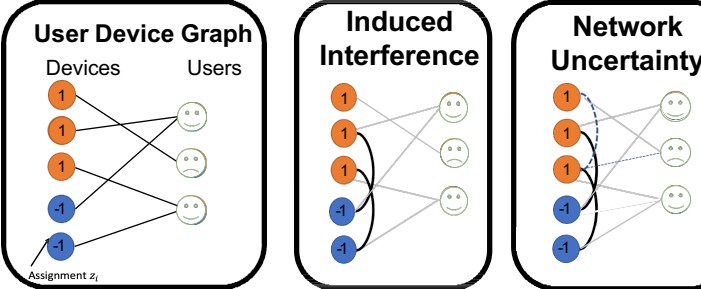

Figure 1: The user device graph presents the connections between the set of users and devices (Left). Treatments $Z_i \in \{-1, 1\}$ applied on a device, exposes the user of the device to the corresponding experience or ad. The outcomes depend on the total exposure a user has had to the treatment. As such the outcome at a device unit $i$ now depends on the assignment of other devices $j$, which induces an interference graph between the devices (Middle). Under uncertain information the induced interference graph has potentially extra (dashed) or fewer edges (Right).

level outcomes are affected by treatments at other units/devices with the same user and not by those of other users, *an assumption can reasonably be made concerning the partial information about the device-user pairings, represented by a structure called 'the device graph'*. Partial information about the device graph be obtained, for instance, from devices with enabled cookies, from geolocation based on IP addresses or from an identity linking model (Sinha et al., 2014; Saha Roy et al., 2015).

In this work, we explore the problem of estimating the *global average treatment effect* (GATE) in the identify fragmentation setting, *under the assumption that interference comes only from devices that share the same user and that, for each user, a superset of their devices is known.* We formalize this problem as treatment effect estimation with interference, where the interference structure is based on the 'device neighborhood' i.e. the set of devices which share a user. We argue that the GATE is identifiable under reasonable assumptions. Finally, we propose a new VAE-based procedure that results in estimators that are superior to existing ones, as demonstrated through extensive experiments on both simulated and real data.

## 2 RELATED WORK

### 2.1 NETWORK INTERFERENCE

Network interference is a well studied topic in causal inference literature, with a variety of methods proposed for the problem. Existing works in in this area incorporate various sets of assumptions to provide an estimate of treatment effects. A common approach is the exposure mapping framework which allows defines a degree of "belonging" of a unit to either the treatment or control group (Aronow et al., 2017; Auerbach & Tabord-Meehan, 2021; Li et al., 2021; Viviano, 2020). A common assumption is that the network effect is linear with respect to a known functional of the neighbour treatments(Basse & Airoldi, 2018; Cai et al., 2015; Chin, 2019; Gui et al., 2015; Toulis & Kao, 2013; Eckles et al., 2017; Sussman & Airoldi, 2017). A limitation of these approaches is that they require complete knowledge of the network structure. Similar to these proposals, our approach also relies on imposing an exposure-based structure to the form of interference, however we can also handle GLM-like outcomes as well an incomplete knowledge of the network.

Treatment effect estimation with unknown network interference has also been studied with the seminal work of Hudgens & Halloran (2008). The key insight behind these works is that if the network can be broken into clusters, then one can perform treatment effect estimation without the full knowledge of the interference structure withing the clusters. Other works such as Auerbach & Tabord-Meehan (2021); Bhattacharya et al. (2020); Liu & Hudgens (2014); Tchetgen & VanderWeele (2012); VanderWeele et al. (2014) have extended this idea further. Often the bias of these estimators depends on the the number of edges between the clusters, which has led to optimization-based methods for constructing clusters (Eckles et al., 2017; Gui et al., 2015). However, this still requires information about the clusters, and is not applicable if multiple clusters of the required type do not

exist. Finally, there are methods, which under restrictive assumptions, use SUTVA based estimates for one-sided hypothesis tests for treatment effect under interference (Choi, 2014; Athey & Wager, 2019; Lazzati, 2015).

**Estimation without any side information**: Recently, some methods have been proposed based on multiple measurements which can address the issue of interference(Shankar et al., 2023b; Cortez et al., 2022; Yu et al., 2022) without any further knowledge. However, such methods assume stationarity i.e. the outcomes do not vary between the trials. This simplifies GATE estimation by providing access to both the factual and counterfactual outcome. However, such a model is unrealistic for our motivating use case of continuous optimization. Furthermore, in the more general settings, conducting multiple trials can be difficult, if not impossible, in itself (Shankar et al., 2023a). As such, we aim to develop a method which can work with only a single trial and/or observational data from an existing test.

## 2.2 ESTIMATION WITH NOISY DATA

Parameter estimation with measurement noise is a well studied problem in causal inference (Wickens, 1972; Frost, 1979). Many methods and heuristics have been proposed for estimation of treatment effect (Carroll et al., 2006; Schennach, 2016; Ogburn & Vanderweele, 2013; Lockwood & McCaffrey, 2016). Yi et al. (2021) provides an overview of recent literature on the bias introduced by measurement error on causal estimation. Earlier works have focused on qualitative analysis by encoding assumptions of the error mechanism into a causal graph Hernán & Robins (2021), outcome Shu & Yi (2019), confounders Pearl (2012); Miles et al. (2018) and mediators Valeri & Vanderweele (2014).

Noisy covariates or proxy variables are not generally sufficient to identify causal effects (Kuroki & Pearl, 2014). As such works such as Kuroki & Pearl (2014); Miao et al. (2018); Shpitser et al. (2021); Dukes et al. (2021); Ying et al. (2021); Guo et al. (2022) have focused on identifying criteria for treatment effect estimation with noisy measurements with confounding variables.

Methods based on assuming knowledge of the error model are also common (Gustafson, 2003; Shpitser et al., 2021; Fang et al., 2023). Consequently, other methods for estimating causal effects also exist relying upon additional information such as repeated measurements (Shankar et al., 2023b; Cortez et al., 2022), instrumental variables (Zhu et al., 2022; Tchetgen et al., 2020) or a gold standard sample of measurements (Shankar et al., 2023a). A few works have also tried to study causal inference with measurement errors and no side information Miles et al. (2018); Pöllänen & Marttinen (2023). Other works have focused on partial identification of treatment effects (Zhao et al., 2017; Yadlowsky et al., 2018; Zhang & Bareinboim, 2021; Yin et al., 2021; Guo et al., 2022), sensitivity analysis (Imbens, 2003; Veitch & Zaveri, 2020; Dorie et al., 2016). Our work differs from these lines of work, as they usually focus on noisy measurements of unknown confounders or covariates, whereas our focus is on unknown network interference.

## 3 NOTATION

We are given a population of $n$ devices. Let $\boldsymbol{Z}$ be the treatment assignment vector of the entire population and let $\mathcal{Z}$ denote the treatments' space, e.g., for binary treatments $\mathcal{Z} = \{-1, 1\}^n$. We use the Neyman potential outcome framework (Neyman, 1923; Rubin, 1974), and denote by $Y_i(\boldsymbol{z})$ the potential outcome for each $\boldsymbol{z} \in \mathcal{Z}$. We can make observations at only the device level, these observations are denoted as $Y_i$ for device $i$. Note that the devices might have a common user, as presented in Figure 1. We assume that the outcome is determined by the user action, and hence the potential outcome at a device $i$ need not depend only on its own treatment assignment but also other treatments allocated to the user's devices. This is a violation of the SUTVA assumption (Cox, 1958; Hudgens & Halloran, 2008); and is commonly called interference or spillover.

The user-device graph induces a dependence between device level outcomes. This dependence can also be represented as a device-level graph (Figure 1(Middle)), where each node represents a device and the presence of an edge indicates a common user between the device pair. The underlying graph is given by its adjacency matrix $\boldsymbol{A} \in \mathbb{R}^{n \times n}$, with $A_{ij} = 1$ only if an edge exists between devices $i$ and $j$, and by convention $A_{ii} = 1$. Let $\mathcal{N}_i(\boldsymbol{A}) = \{j : A_{ij} = 1\}$ be the set of *neighbors* of device $i$. Since we assume the underlying graph is fixed, we will use $\mathcal{N}_i(\boldsymbol{A})$ and $\mathcal{N}_i$ interchangeably. We assume that the outcomes depend on the treatments received by a user (i.e. SUTVA holds at the user level). This implies that the interference at a device is limited to its neighbours in the graph.

$$\text{User Level SUTVA: } \forall \, \boldsymbol{z}, \boldsymbol{z}' \in \mathcal{Z} \text{ s.t.} \quad z_i = z_i' \text{ and } z_j = z_j' \; \forall j \in \mathcal{N}_i : \quad Y_i(\boldsymbol{z}) = Y_i(\boldsymbol{z}'). \quad \textbf{(A1)}$$

We will assume that the experimental design is a randomized Bernoulli design i.e. each device $i$ gets allotted the treatment $z_i = 1$ independently with probability $p \in (0, 1)$. This is analogous to the standard randomization and positivity assumption in causal inference, and is equivalent if one assumes the exposure map $Y_i(\boldsymbol{z})$ only depends on $z_i$.

The desired causal effect is the mean difference between the outcomes when $\boldsymbol{z} = \vec{1}$ i.e. $z_i = 1 \, \forall i$ and when $\boldsymbol{z} = \vec{0}$ i.e. $z_i = -1 \, \forall i$. Under the aforementioned notations, this causal effect is given by:

$$\tau(\vec{1}, \vec{0}) = \frac{1}{n} \sum_{i=1}^{n} Y_i(\vec{1}) - \frac{1}{n} \sum_{i=1}^{n} Y_i(\vec{0}) \tag{1}$$

If the true graph $\boldsymbol{A}$ is known, under certain assumptions one can estimate the above treatment effect (Hudgens & Halloran, 2008; Halloran & Hudgens, 2016). However, in our problem setting, knowledge of the true graph would imply knowing which devices belong to the same user. As such we cannot assume, that $\boldsymbol{A}$ is known. However, we have access to some information about $\boldsymbol{A}$. In our use case of online experimentation, this information can come from those devices where the user has given cookie permissions, or from covariate information like geography or IP addresses, or from some existing model user for identity linking (Sinha et al., 2014).

Finally, we assume access to a model $\mathcal{M}$ which provides information on $\boldsymbol{A}$. Specifically, we assume that the $\mathcal{M}$ can be queried for any device $i$ to get a predicted (or assumed) neighbours of a device (see Figure 1 (Right)). We will denote this neighbourhood by $\mathcal{M}(i)$.

Our primary focus revolves around estimating the Generalized Average Treatment Effect (GATE) under the previously outlined scenario, where there exists a degree of uncertainty concerning the network structure. Before we delve further into the method we provide a brief explanation of commonly used estimators and their problems for our problem setting.

**Inverse Propensity/Horvitz-Thompson Estimate**    If the graph is known and when all treatment decisions are iid Bernoulli variables with probability p: one can use the classic Horvitz Thompson estimator as follows:

$$\frac{1}{n} \sum_i Y_i \left( \frac{\prod_{j \in \mathcal{N}_i} z_j}{\prod_{j \in \mathcal{N}_i} p} - \frac{\prod_{j \in \mathcal{N}_i} (1 - z_j)}{\prod_{j \in \mathcal{N}_i} (1 - p)} \right) = \frac{1}{n} \sum_i Y_i \left( \prod_{j \in \mathcal{N}_i} \frac{z_j}{p} - \prod_{j \in \mathcal{N}_i} \frac{(1 - z_j)}{(1 - p)} \right)$$

This inverse propensity estimators (and its derivatives) do not require any further assumption other than randomization and positivity to be unbiased. However, on inspection, one can see that this estimator ignores any units for which all neighbours are not in control or treatment groups. This results in extremely high variance, as most data samples are ignored. Moreover, if the number of neighbours is large, then this estimate may not even have a meaning, as there may not exist units for which all the neighbours are in control or treatment groups. This is particularly troublesome for our application as uncertainty in the graph means accounting for more possible units which interfere with a given unit, and including such units adds to the estimation issue of HT-estimators.

**SUTVA Estimate**    The SUTVA estimate is given by

$$\hat{\tau}_{SUTVA} = \bar{Y}^1 - \bar{Y}^{-1} = \frac{\sum Y_i \mathbb{I}[Z_i = 1]}{\sum \mathbb{I}[Z_i = 1]} - \frac{\sum Y_i \mathbb{I}[Z_i = -1]}{\sum \mathbb{I}[Z_i = -1]}$$

where $\bar{Y}^{-1/1}$ are the average of observed outcomes for units where $Z_i = -1/1$ respectively. Since it is the difference in means of control and treatment groups, it is also called the difference in mean/ DM estimator. This estimator while quite efficient and practical, requires the SUTVA assumption to be unbiased. As such these estimators can be misleading when it comes to our scenario.

## 4 METHOD

### 4.1 MODEL AND ASSUMPTIONS

Randomized experiments with interference (even with neighbourhood interference) can be difficult to analyze since the number of potential outcome functions grows exponentially: $2^{\mathcal{N}_\rangle}$ for unit $i$; unlike the SUTVA case where one has only two outcomes. As such the literature around network interference restricts the space of potential outcome functions in order to do meaningful inference. One common approach is the exposure function (or exposure mapping) approach. Under this model one uses exposure variables which are functions from the discrete combinatorial space $\{-1, 1\}^{\mathcal{N}_i} \to \mathbb{R}^d$. One posits that the outcome $Y_i$ depends on the treatment $z$ only via the exposure variable $e_i$ (Hudgens & Halloran, 2008; Aral & Walker, 2011; Aronow et al., 2017; Brennan et al., 2022). We will abuse notation, and often use $e_i$ instead of the functional notation $e_i(z)$.

We too consider an exposure model; specifically we assume an outcome model of the form

$$Y_i(z, x_i) = \underbrace{\mu_Y(z, x_i)}_{\mathbb{E}[Y|Z=z, X_i=x_i]} + \epsilon = c_0(x_i) + c_1(x_i)z_i + g(w(x_i)^T e_i(z, x_i)) + \epsilon \qquad \textbf{(A2)}$$

where $\epsilon$ is mean zero noise, and $x_i$ are the covariates at unit $i$. Assumption **A2** as stated is very generic, since the exposure function itself can be arbitrary. For meaningful inference, one often invokes a specific parametric form for the exposure function. A common example is an exposure represented as the (weighted) proportion of neighboring units that have received treatment (Eckles et al., 2017; Toulis & Kao, 2013). Alternatively, it could involve the count of neighboring units that have undergone treatment (Ugander et al., 2013). We will assume an additive vector exposure function along with some other standard assumptions (stated below) from treatment effect literature (Pearl, 2009).

$$\text{Additive Exposure: } e_i = \sum_{j \in \mathcal{N}_i} \phi(z_j, X_i) \quad \textbf{(A3)}$$

$$\text{Network Ignorability: } Y(z) \perp\!\!\!\perp Z \,\forall z \quad \textbf{(A4)}$$

$$\text{Positivity: } P(z|X) > 0 \,\forall z \quad \textbf{(A5)}$$

$$\text{Consistency: } Y_i = Y_i(z) \text{ if } Z = z \quad \textbf{(A6)}$$

$$\text{Neighbourhood Superset: } \mathcal{M}(i) \supseteq \mathcal{N}_i \quad \textbf{(A7)}$$

Since $\phi$ in Assumption (**A3**) depends on the individual covariates, this assumption supports unit-level observed heterogeneity. We can also include the covariates $x_j$ of the neighbouring units as well in $\phi$ but ignore this for simplicity. Further $\phi$ can be a vector function instead of scalar, and so **A3** can support all set function of neighbourhood treatments (Braun & Griebel, 2009). Moreover it also supports other common assumptions such as those in (Toulis & Kao, 2013; Eckles et al., 2017; Pouget-Abadie et al., 2019)

**Remark 1.** *A7 can seem to be a strong assumption. However, in many applications, it is not difficult to satisfy this assumption. As a simple example, consider all devices which share a geographic location, with a given device $i$. This is very likely to be a superset of all devices that share a user with $i$. Furthermore, in practice, device-linking methods are used to identify neighbours based on confidence scores. These methods can usually be adapted to obtain a superset of neighbours with high probability ( by including even low confidence nodes as neighbours).*

### 4.2 MODEL TRAINING

We propose using a latent variable model to infer the treatment effect. The dependence between various variables is depicted in Figure 2. We denote by $E$ the true exposure which is the key latent variable of the model. $\tilde{E}$ is the exposure as implied by $\mathcal{M}$, which is our uncertain representation of the underlying device graph. The key difference between this and a standard exposure based causal model, is that in the latter the true exposure $E$ is observed whereas in our model it is unobserved. Instead of $E$ we observe the noise corrupted value $\tilde{E}$.

**Remark 2.** *Note that the true exposure $E$ depends on the actual neighbourhood $\mathcal{N}_i$, while the observed exposure $\tilde{E}$ depends on the assumed neighbourhoods $\mathcal{M}(i)$.*

Fundamentally, this is a discrete problem as $Z$ is a binary assignment of treatments at individual devices. However, since training such models is computationally intensive , we use a variational autoencoder (VAE) (Kingma & Welling, 2013; Kingma et al., 2019) based approximate training. In the appendix we argue why this procedure is analogous to the learning method suggested in Schennach & Hu (2013).

We posit a generative model for the joint distribution $p_\theta(\tilde{E}, E, Y|X, Z)$ which factorizes as $p_\theta(Y|E, X)p(\tilde{E}|E)p(E|Z)$. For the outcome distribution $Y$ we posit a GLM style model which parameterizes $\mathbb{E}[Y|Z = z, X = x]$ from **A2** in terms of a neural network i.e. we use a neural network for each of the function $c_0, c_1, g, w$ in **A2**. For the $p(\tilde{E}|E)$ we use a Gaussian model. If $|\mathcal{M}(i)| >> \mathcal{N}_i$, by law of large numbers this is a good approximation for the error. Finally $p(Z|X)$ is just the allocation mechanism which is exactly known to us as the experimenter.

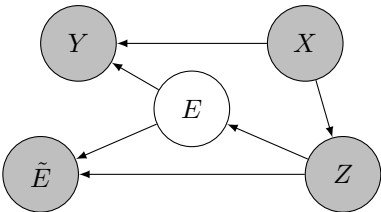

Figure 2: Graphical model depicting relationships between different variables for our model. Observed variables $\tilde{E}$ (noisy exposure), $Y$ (effect/outcome), and $X$ (covariates), $Z$ (treatment allocation) are shaded to distinguish them from the hidden variable $E$ (true treatment).

To use VAE style learning one needs to specify a posterior $q_\phi$ for the latent variable. For this we use a Gaussian variational approximation with both mean and variance parameterized. Specifically we use a $q$ of the form $N(e|\mu_q(\tilde{e}, x, y; \phi), \sigma_q(\tilde{e}, x, y; \phi))$. As our objective function, we use the $K$-sample importance weighted ELBO $\mathcal{L}_K$ Burda et al. (2016), which is a lower bound for the conditional log-likelihood $p_\theta(x, y|z)$:

$$\mathcal{L}_K = \sum_{i=1}^{N} \mathbb{E}\left[\log \frac{1}{K} \sum_{j=1}^{K} w_{i,j}\right] \leq \sum_{i=1}^{N} \log \mathbb{E}\left[\frac{1}{K} \sum_{j=1}^{K} w_{i,j}\right] = \log p_\theta \qquad (2)$$

where $w_{i,j} = p_\theta(\tilde{e}_i^*, z_{i,j}, x_i, y_i)/q_\phi(e_{i,j}|\tilde{e}_i, x_i, y_i)$ are importance weights, and the expectation is respect to $q_\phi$. To reduce training variance we use the recent DReG estimator (Tucker et al., 2018). Once the model $p_\theta$ has been trained, one can obtain estimates of the mean outcomes $\mu_Y(\boldsymbol{z}, x_i)$ using $p_\theta(Y|E, X)$. By plugging the estimated outcomes into Equation 1, we get our estimate $\hat{\tau}$ [2].

**Remark 3.** *While the probability distribution can be arbitrarily parameterized with neural networks, all the neural networks used in our experiments, are MLPs with one hidden layer and ReLU activation.*

### 4.3 IDENTIFIABILITY

A key concern in causal inference, is the identifiability of the desired estimand, as otherwise there is no justification for the estimated value to correspond to the ground truth. Next, we discuss the identifiability of the treatment effect in the aforementioned scenario. The identifiability of treatment effect in our model is related to results in Schennach & Hu (2013). We summarize the crux of the argument below, while deferring the details to Appendix A

**Proposition 1.** *Under Assumptions **A1-7** and certain technical conditions on the function $\mu_Y$, the conditional mean function $\mathbb{E}[Y|Z = z, X = x] = \mu_Y(x, z)$ is identifiable.*

Under **A2,4-6**, the problem of treatment effect estimation becomes a model fitting problem. Specifically, if the exposures $e_i$ are known, one can conduct a regression of the observed outcomes $Y_i$ on the exposures $e_i$ and covariates $X_i$ to estimate the population-level mean potential outcomes functions, denoted as $\mu_Y$. Once we estimate the mean potential outcomes, we can obtain the treatment effect $\tau$ by plugging in these estimates into Equation 1.

When the graph $A$ is exactly known, one can compute the exposures $e_i$ using Assumption **A3**. However, since in our problem, the graph is unknown, obtaining $e_i$ is not possible. To address this obstacle, we reframe the inference problem in our scenario as a regression with a measurement

---

[2]Refer to Appendix for more details

error problem. Observe that the exposure $e_i$ under the assumed graph $\mathcal{M}$ is given by $e_i(\mathcal{M}) = \sum_{j \in \mathcal{M}(i)} \phi(z_j, X_i)$. Due to **A7** $e_i(\mathcal{M})$ can be decomposed as $e_i(\mathcal{N}_i) + \Delta e_i$, where $\Delta e_i$ is an independent error term. Thus we can use $e_i(\mathcal{M})$ as noisy estimates of $e_i(\mathcal{N}_i)$.

Next, we argue the identifiability of the above regression task. Schennach & Hu (2013) provide conditions under which models of the form:

$$Y = \mu_Y(E) + \Delta Y; \quad \tilde{E} = E + \Delta E \ \ \Delta E \perp\!\!\!\perp E$$

can be identified from only the joint observations of $Y, \tilde{E}$. We show that the under assumptions **A1-6**, the conditions required for the identifiability results in Schennach & Hu (2013) are satisfied, thus making our model identifiable [3]. A detailed discussion is provided in the Appendix.

**Remark 4.** *This result does not apply when $\mathcal{M}(i) \subset \mathcal{N}_i$ because then the error term $\Delta \epsilon_i = \epsilon_i(\mathcal{M}) - \epsilon_i(\mathcal{N}_i)$ is no longer independent of the true exposure $\epsilon_i(\mathcal{N}_i)$. In that case, the our approach becomes equivalent to regression with endogenous covariate error, which requires additional information ().*

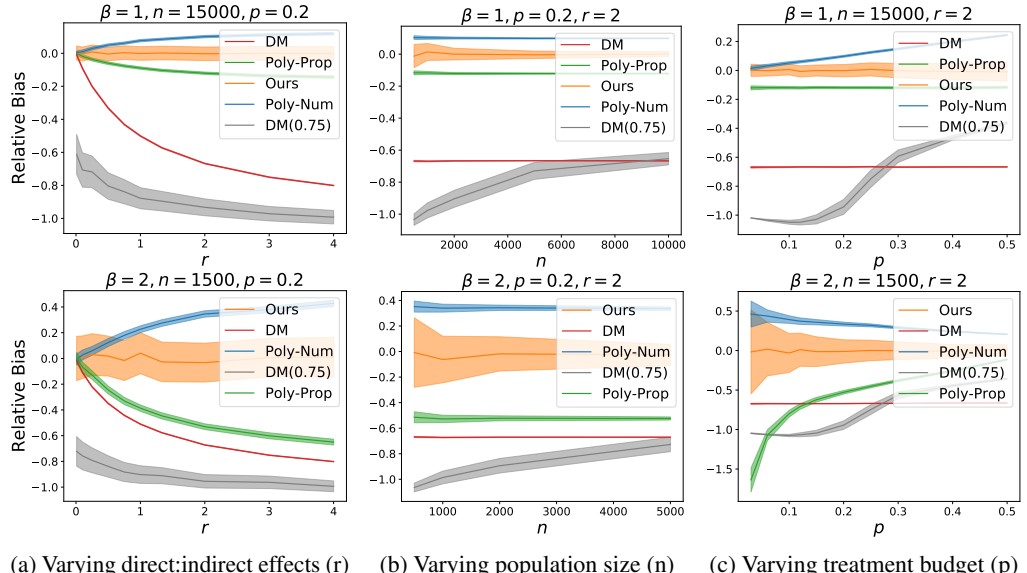

(a) Varying direct:indirect effects (r)    (b) Varying population size (n)    (c) Varying treatment budget (p)

Figure 3: Plots visualizing the performance of various GATE estimators under Bernoulli design on Erdos-Renyi networks for both linear and quadratic potential outcomes models. The lines represent the empirical relative bias i.e. $\frac{\hat{\tau} - \tau}{\tau}$ of the estimators across different settings, with the shaded width corresponding to the experimental standard error.

## 5 EXPERIMENTS

### 5.1 SYNTHETIC GRAPHS

In this section, we first experimentally demonstrate the validity of our approach by experimenting with synthetic data obtained from a model which satisfies our assumptions exactly. For this we experiment with synthetically generated Erdos-Renyi graphs to compare the performance of our estimator with other estimators. We simulate 100 different random graphs and run repeated experiments on this graph with random treatment assignments. We sample covariates $X$ independently from a multivariate normal distribution and consider a polynomial family of outcome models. Specifically the outcomes are simulated from the following equation $Y_i(z, X_i) = c_0(X_i) + g(w(X_i)^T \sum_{j \in \mathcal{N}_i} \phi_{i,j}(z_j)) + \epsilon$ where $g$ is a polynomial function of order $\beta$ and $\epsilon$ is mean 0 error. Similar to Cortez et al. (2022), we experiment with the linear $\beta = 1$ and quadratic $\beta = 2$ setting. For each experiment, we varied the treatment probability $p$, the size of the graphs $n$ to assess the efficacy of estimation across different ranges of parameters and the strength of interference $r$. Following Cortez et al. (2022), the strength of

---

[3] The primary restriction is that g should not be of the form $g(z) = a + b \ln(\exp(cz) + d)$

interference is measured as the ratio of norms of the self-influence $\phi_{i,i}$ and average cross-influences $\phi_{i,j}$ i.e. $r = \frac{1}{n} \sum_i \frac{\sum_{j \in \mathcal{N}_i \setminus i} |\phi_{i,j}|}{|\phi_{i,i}||\mathcal{N}_i|}$

**Baselines**   In our evaluation, we gauge the effectiveness of our proposed method by benchmarking it against commonly employed estimators such as polynomial regression (Poly), difference-in-means (DM) estimators. Since the polynomial regression model needs exact neighbourhoods, we use them in an oracle setting i.e. they have access to the true device graph. [4]

The results are presented in Figure 3. From the figure it is clear that our model produces unbiased estimates in this case. On the other hand, all other methods produce highly biased estimates. Note that in Figure 3a, when $r = 0$, there is no interference, and hence most estimators are unbiased. However, when interference increases these methods clearly show strong bias. Secondly, for a given interference strength, our method shows consistency in the form of decreasing variance with increasing number of nodes. Finally, the variance of our method reduces as the treatment probability $p$ increases to 0.5.

## 5.2   AIRBNB SIMULATIONS

Next, we conduct simulation from a model designed from the AirBnB vacation rentals domain Li et al. (2022). The original model is a simulator for rental listings and their bookings for a two-sided marketplace. Contrary to the previous experiments, the outcomes here do not follow an explicit exposure mapping. We adapt this simulator for our purposes, replacing customers with devices and listings with users. The measured outcome $Y_i$ is 1 iff there is a click on device $i$. A user watches ads on a randomly chosen subset of its devices, and chooses to click on the ad on only one device, leading to interference between outcomes. This simulation works uses a type matching model where if the device and person have the same type, the probability of watching an ad on that device is higher. The treatment scales the probability of seeing an ad by the parameter $\alpha$. This is a good testbed for testing robustness of our model, since like in the real-world, exposure models are only our best approximations to the unknown and complex actual interference function. We perform simulations with protocol specified in Brennan et al. (2022) [5].

**Baselines**   As baselines in this experiment, we use the SUTVA/DM estimator, an exposure model with oracle graph i.e. one where the exact graph is known (labelled Exp), and a Horvitz-Thompson estimator with oracle graph (labelled HT). The Exp model is same as the one used in Brennan et al. (2022), while the HT estimator is the one described in Section 3. The performance of different estimators is shown in Figure 4.

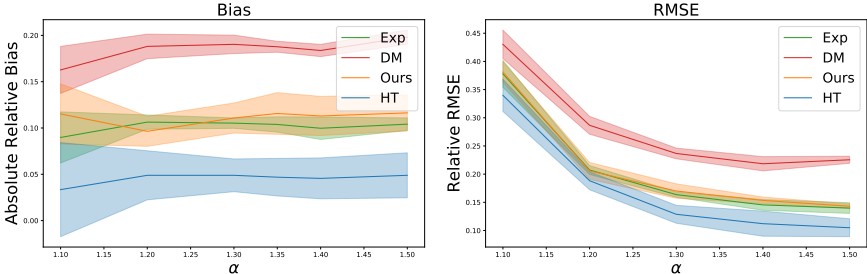

Figure 4: Visualization of performance of different GATE estimators on the airbnb simulator. The lines represent a) absolute relative bias $|\frac{\hat{\tau}-\tau}{\tau}|$ and b) relative RMSE of various algorithms as the indirect treatment effect $\alpha$ increases. Bands capture the standard deviation over 500 trials.

Since the exposure model can only partly model the actual outcomes, in this case, bias is not zero. On the other hand, the Oracle HT estimator (which makes no exposure assumptions) gives unbiased though higher variance estimates. The model is Oracle in using the exact interference graph. A different model is the Oracle Exposure (Exp) model which used the true graph to compute the

---

[4]Due to incorporating large neighbourhoods (with upto 100 extraneous nodes), Horwitz-Thompson type estimators failed to yield non-meaningful results in any trial.

[5]Details in Appendix

exposure. From the result it is also clear that our approach works as well as the Oracle Exposure model. Furthermore, even on the MSE metric our model performs comparably to the Exp model. These results suggest that our method is robust even when the true potential outcome does not obey the assumed exposure mapping.

## 5.3    Effect of Network Uncertainty

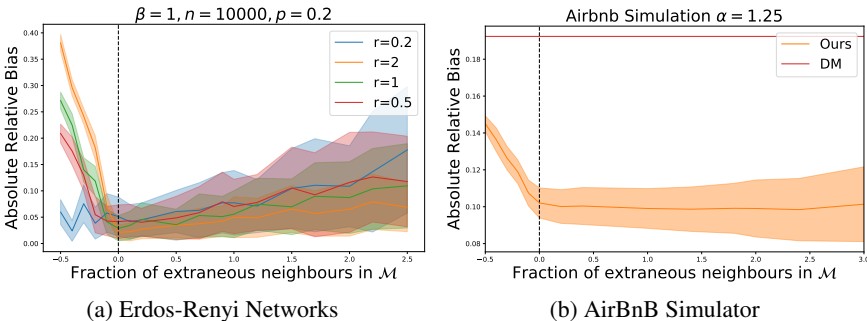

(a) Erdos-Renyi Networks                    (b) AirBnB Simulator

Figure 5: Impact of neighbourhood sizes on the absolute relative bias i.e. $|\frac{\hat{\tau}-\tau}{\tau}|$ GATE estimation. Negative fraction of neighbours indicate the case when $\mathcal{M}(i) \subset \mathcal{N}_i$ i.e. we missed pertinent neighbours. The bias tends to be high when gives small neighbourhoods, as they miss pertinent edges. As the neighbourhood sizes increase, the bias reduces, but the uncertainty widens.

Next we examine the impact of the neighborhood accuracy $\mathcal{M}(i)$ in estimation. We experiment with Erdos-Renyi graphs as well as with the AirBnB Model. For these experiments, we fix a single graph, and compute the treatment effect estimate from our method as we change the assumed neighbourhoods $\mathcal{M}(i)$. In Figure 5a, we preset the relative ratio between the estimated and true treatment effects as varying proportions of edges are either added or omitted by $\mathcal{M}(i)$. To maintain simplicity, we maintain uniform $\mathcal{M}(i)$ sizes across all nodes, employing the average number of missed or added edges as the metric along the x-axis. Figure 5b presents the same experiment within the context of the Airbnb simulator. We observe a similar trend in both experiments: when $\mathcal{M}(i) \supseteq \mathcal{N}_i$ holds true for all nodes $i$, our approach can offer an lower bias estimate of the treatment effect. Nonetheless, as the number of extraneous nodes within $\mathcal{M}(i)$ grows, so does the uncertainty in estimation. Conversely, if $\mathcal{M}(i)$ neglects a pertinent node, it may introduce greater bias into the estimation process. This manifests within our results, where the model predictions initially exhibit strong bias. However, as neighborhood sizes expand, bias diminishes while variance increases.

## 5.4    Application: Assessing Power Plant Emissions Controls

We use our approach to estimate the effect of pollution reduction technologies on ambient ozone levels. As ambient pollution is heavily influenced by spatially adjacent sources of pollution, adjusting for interference is important. DM estimators in this case often underestimate the impact in these scenarios. We work with a public dataset on 473 power generation facilities in USA used in Papadogeorgou et al. (2019). We use the DM, Poly and Exp estimators as baselines of which the latter two need exact neighbourhoods. For our method we will not use coordinate information for identifying neighbourhoods and instead uses groupings based on census divisions. The results (Figure: 6) show that our method provides comparable estimates to other oracle estimators.

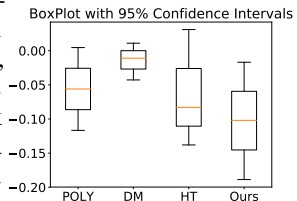

Figure 6: GATE on ambient ozone levels of adopting of SCR/SNCR technologies

## 6    Conclusion

Identity fragmentation is an increasingly relevant problem in online A/B testing. Our work provides a method to estimate GATE under a relaxed assumption of having knowledge only about the super-set of the identities that belong to the user. This relaxed assumption can be practically far more feasible than requiring the exact network. With both theoretical and experimental analysis, we established the efficacy of our estimator(s) under this assumption.

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
