## A   Estimation and Identifiability

**Proposition 1.** *If the neighbourhood proposed by $\mathcal{M}$ i.e. $\mathcal{M}(i)$ always contains the true neighbourhood $\mathcal{N}_i$, and is sufficiently larger than $\mathcal{N}_i$, then under the exposure assumption we can treat $\Delta Z$ as approximately gaussian.*

*Proof.* Under **A7** we can rewrite the exposure under $\mathcal{M}$ as:

$$e_i(\mathcal{M}) = \sum_{j \in \mathcal{M}(i)} \phi(z_j, X_i) = \sum_{j \in \mathcal{M}(i) \cap \mathcal{N}_i} \phi(z_j, X_i) + \sum_{j \in \mathcal{M}(i) - \mathcal{N}_i} \phi(z_j, X_i)$$

Now, since allocation of device level treatments are independent, $Z_i \perp\!\!\!\perp Z_j$, as well as its independent of $X_i$, the individual exposure terms $\phi(Z_j, X_i) \perp\!\!\!\perp Z_i$ for any $i \in \mathcal{M}(i) - \mathcal{N}_i$. If $|\mathcal{M}(i)| >> |\mathcal{N}_i|\phi(z_j, X_i)$, then the central limit theorem implies that the sum is approximately $\sum_{j \in \mathcal{M}(i) - \mathcal{N}_i} \phi(z_j, X_i)$ as $N(\bar{\phi}, |\mathcal{M}(i) - \mathcal{N}_i|Var(\phi)) \approx N(\bar{\phi}, |\mathcal{M}(i)|Var(\phi))$ □

**Proposition 2.** *Our model is identifiable if 1) $\forall x, \mu_Y(x, z)$ is continuously differentiable everywhere as a function of z, and 2) $\forall x, \partial_z \mu_Y(x, z) \neq 0$*

Before arguing the previous proposition, we first state Theorem 1 from Schennach & Hu (2013). Our presentation broadly follows that of Pöllänen & Marttinen (2023).

**Theorem 1 from Schennach & Hu (2013)**: Let $y$, $z$, $z^*$, $\Delta z$, $\Delta y$ be scalar real-valued random variables related through

$$y = g(z^*) + \Delta y \tag{3}$$
$$z = z^* + \Delta z, \tag{4}$$

and $y, z$ are observed while all remaining variables are not and satisfy the following assumptions:

**Assumption 1.** The variables $z^*$, $\Delta z$, $\Delta y$, are mutually independent, $\mathbb{E}[\Delta z] = 0$, and $E[\Delta y] = 0$ (with $\mathbb{E}[|\Delta z|] < \infty$ and $\mathbb{E}[|\Delta y|] < \infty$).

**Assumption 2.** $\mathbb{E}[e^{i\xi\Delta z}]$ and $\mathbb{E}[e^{i\gamma\Delta y}]$ do not vanish for any $\xi, \gamma \in \mathbb{R}$, where $i = \sqrt{-1}$.

**Assumption 3.** (i) $\mathbb{E}[e^{i\xi z^*}] \neq 0$ for all $\xi$ in a dense subset of $\mathbb{R}$ and (ii) $\mathbb{E}[e^{i\gamma g(z^*)}] \neq 0$ for all $\gamma$ in a dense subset of $\mathbb{R}$ (which may be different than in (i)).

**Assumption 4.** The distribution of $z^*$ admits a uniformly bounded density $f_{z^*}(z^*)$ with respect to the Lebesgue measure that is supported on an interval (which may be infinite).

**Assumption 5.** The regression function $g(z^*)$ is continuously differentiable over the interior of the support of $z^*$.

**Assumption 6.** $g'(z^*)! = 0$ almost everywhere, and $f_{z^*}(z^*)$ is continuous and nonvanishing

**Theorem 1.** *Let Assumptions 1-6 hold. Then the following holds:*

1. *$g(z^*)$ is not of the form*

$$g(z^*) = a + b\ln(e^{cx^*} + d) \tag{5}$$

   *for some constants $a, b, c, d \in \mathbb{R}$. Then, $f_{z^*}(z^*)$ and $g(z^*)$ (over the support of $f_{z^*}(z^*)$) and the distributions of $\Delta z$ and $\Delta y$ are identified.*

2. *If $g(z^*)$ is of the form (5) then, neither $f_{z^*}(z^*)$ nor $g(z^*)$ in Model 1 are identified iff $z^*$ has a density of the form*

$$f_{z^*}(z^*) = A \exp(-Be^{Cx^*} + CDx^*)(e^{Cx^*} + E)^{-F}, \tag{6}$$

   *with $c \in \mathbb{R}$, $A, B, D, E, F \in \mathbb{R}^+$*

Next, we argue how Theorem 1 implies Proposition 2.

Consider the conditional versions of our, i.e. consider the restricted version where the covariates $X$ have been fixed. It is clear from Proposition 1 and Assumption **A2** that Equations (3) and (4) are satisfied for such a model. Assumption 1 of Theorem 1 also follows from Proposition 1 and Assumption **A2**.

Assumptions 2,3 are technical conditions satisfied by most distributions ( including Gaussian, Uniform and exponential family distributions). Assumption 4 is satisfied because $\tilde{E}|E$ is approximately normal. Furthermore it will also hold for a variety of bounded continuous distributions. Assumption 5,6 hold from the assumption on $\mu_Y$ stated in the proposition. With the assumptions of Theorem 1 satisfied, the conditional mean function $\mathbb{E}[Y|Z.X = x]$ are identified based on the two conditions Theorem 1 except for when $\mu_Y(x, z^*)$ might be of the form $a + b\ln(e^{cz^*} + d)$.

Since the conditional means $\mu_Y(Z, X = x)$ is identifiable for all $x$, the overall function $\mu_Y(Z, X)$ is also identified.

## A.1 RELATION OF OUR MODEL TO SCHENNACH & HU (2013) METHOD

Schennach & Hu (2013) proposed estimating the function $g$ in Equation 3 through the following optimization.

$$g = \arg\max_g \max_{f1, f2, f3} \ln \int f1(y - g(z^*)) f2(z - z^*) f3(z^*) dz* \tag{7}$$

where $f1, f2, f3$ are restricted to be probability densities. This method is effectively maximizing the log-likelihood of the observed data under a latent variable framework. The latent variable, denoted as $z^*$, is integrated out within the objective which is a normalized density. Comparing this equation with our Equation 2, it becomes apparent that these methods are related. Specifically, the log-likelihood in Equation 2; can be obtained from Equation 7 by replacing $z^*$ with $e$ and $z$ by $\tilde{e}$. The two key differences between our objective and that of Schennach & Hu (2013) is a) that our likelihoods model conditioned on covariates $X$, and b) we can use specifics form for the densities $f2, f3$. The first difference is natural as we are fitting conditional models, unlike Schennach & Hu (2013). The choice of specific densities is also not an issue in our scenario. As the experimenter, we already know the data generating density $f3$ function, and by Proposition 1, $f2$ is well approximated by a Gaussian. This such eliminates the need to learn these densities in our approach. Given ideal conditions, such as fully flexible posteriors and exact optimization, our proposed method converges towards the same solution as that obtained by the method of Schennach & Hu (2013).

## A.2 ESTIMATION

Here we describe obtaining the estimate of treatment effect $\hat{\tau}$ from the model learnt in Section 4.2. We note that the variational posterior $q_\phi$ is providing us the estimate of the latent exposures $E$, while the model $p_\theta(Y|E, X)$ is learning the outcome models. Specifically, since $p_\theta(Y|E, X)$ is a GLM-style model, one can directly obtain the mean counterfactual outcome from it. Next these estimated means can be plugged in Equation 1 to get $\hat{\tau}$.

Under **A2**, this computation is further simplified by noting that output of $c_0$ is independent of the treatment $z$. Furthermore, we can see from **A2** that the mean $\mathbb{E}[Y|E, X]$ is direct sum of the output of the networks $c_0, c_1, w$ when provided the corresponding inputs. As such one can directly obtain the treatment effect using the following equation:

$$\hat{\tau} = \frac{1}{n} \sum_i^n \hat{\mu}_Y(\vec{1}, x_i) - \hat{\mu}_Y(\vec{0}, x_i)$$

$$= \frac{1}{n} \sum_i^n \left[ c_1(x_i) + g(w(x_i)^T e_i(\vec{1}, x_i)) \right]$$

Here $c_1, g, w$ etc are neural networks whose parameter was estimated in learning $p_\theta$.

## B  EXPERIMENTAL DETAILS

### B.1  AIRBNB MODEL

The model used in these experiments is a version of the buyer and listing simulator developed by Li et al. (2022). The original model is a simulator for rental listings and their bookings for a two-sided marketplace scenario, with treatments affecting which seller listings are applied to by a buyer.

We adapt this simulator for our purposes, replacing customers with devices and listings with users. Each device and customer have a latent category, and the probability of watching an ad is significantly higher if the user and device category match. We assume that the watches all ads that it has decided to watch, and then with a certain probability clicks on only of the ads it sees. Effectively, there is no temporal component in the treatments. The observed outcome $(Y_i)$ is 1 if device $i$ successfully receives a click on the ad. Since only one click is possible, the more ads are watched by the user, the lesser is the click rate at a single device, leading to interference. In our terminology, the experimental units are devices and the interference units are users. The outcomes at an experimental unit is influenced by other experimental units that are incident on the the same interference unit. Consistent with prior literature (Li et al., 2022; Johari et al., 2022; Brennan et al., 2022), we use a 20 latent type matching model. To match consideration probabilities as mentioned in Brennan et al. (2022) the ad-watching probability under the control assignment is 0.016 if the device and user share the same type. Similarly, the click probability was set to match acceptance probabilities mentioned in Brennan et al. (2022). The treatment tested is a recommendation algorithm, which increases the probability that a user watches an ad on the treated device.

### B.2  SYNTHETIC GRAPHS

The Erdos-Renyi (ER) model is commonly used for analyzing interaction networks in various experimental settings, particularly in the realm of social media (Seshadhri et al., 2012) and epidemic control (Kephart & White, 1992; Wang et al., 2003). In social media platforms, where connections form organically, ER graphs provide a reasonable simulation of how friendships, followerships, or interactions might evolve in an online community (Erdos et al., 1960). Additionally, in the context of epidemic control, ER graphs are valuable for studying disease spread (Wang et al., 2003).

We sample different random Erdos-Renyi Graphs and run repeated experiments on these graphs with randomized bernoulli treatment assignment. The baselines include the POLY(Prop/Num) estimator is a polynomial regression on the exposure as computed by the fraction/number of treated nodes in the neighbourhood. The DM estimator signifies the classic difference in mean/ SUTVA estimator which is is simply the average outcomes on treated vs un-treated units. The ER graphs are made with an expected neighbourhood of size 20. The outcome model is similar to the potential outcomes model as in Cortez et al. (2022):

$$Y_i(\boldsymbol{z}) = c_{i,\emptyset} + \sum_{j \in \mathcal{N}_i} \tilde{c}_{i,1} z_j + \sum_{\ell=2}^{\beta} \left( \frac{\sum_{j \in \mathcal{N}_i} \tilde{c}_{ij,2} z_j}{\sum_{j \in \mathcal{N}_i} \tilde{c}_{ij,2}} \right)^{\ell}, \tag{8}$$

where $i \neq j$, $\tilde{c}_{ij,2} = v_{i,2} |\mathcal{N}_i| / \sum_{k:(k,j) \in E} |\mathcal{N}_k|$. The coefficient $c_{i,\emptyset}, \tilde{c}_{i,1}, v_{i,2}$ are obtained from the device covariates $X_i$.

We also provide the signed relative bias and RMSE plots from these experiments in Figure 7

### B.3  POWER PLANT EMISSION EXPERIMENTS

Selective Catalytic Reduction (SCR) and Selective Non-Catalytic Reduction (SNCR) are effective emission reduction technologies used in industrial settings, and their effectiveness in pollution has been supported in literature (Papadogeorgou et al., 2019). As ambient pollution is heavily influences by spatio-temporally adjacent sources of pollution, interference is a key component in the study of air pollution. We employ the identical dataset as Papadogeorgou et al. (2019) to appraise the effect of SCR/SNCR adoption on ambient ozone levels. This openly accessible dataset encompasses 473 coal or gas-fired power generation facilities in USA. The dataset provides covariate details encompassing power plant characteristics, weather conditions, and demographic information in the surrounding regions. Due to the knowledge of geographical proximity, spatial-interference aware estimation

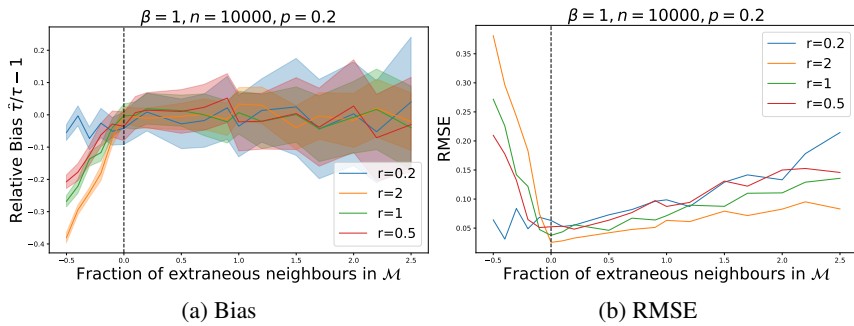

(a) Bias

(b) RMSE

Figure 7: Visualization of the impact of neighbourhood sizes on GATE estimation. Negative fraction of neighbours indicate the case when $\mathcal{M}(i) \subset \mathcal{N}_i$ i.e. we missed pertinent neighbours. The bias tends to be high when gives small neighbourhoods, as they miss pertinent edges. As the neighbourhood sizes increase, the bias reduces, but the uncertainty widens.

methods can be used to provide plausible estimates of the treatment effect (Papadogeorgou et al., 2020). The POLY(Prop/Num) estimator is a polynomial regression on the exposure as computed by the fraction/number of treated nodes in the neighbourhood. The EXP estimator is the augmented inverse propensity estimator used by Papadogeorgou et al. (2020), using a spatial exposure model. DM is the direct difference in mean estimate. For the DM estimate, a unit is considered treated if the nearest power-station adopted pollution reduction measures. The Poly and EXP require oracle neighbourhoods which can be obtained using latitude and longitude information. However, for our method we will not leverage such information for identification of the interfering neighbourhood sites, and instead use clusters based on the 9 census divisions as specified by the US Census Bureau.