# OpenReview forum: "A/B testing under Identity Fragmentation"
_ICLR.cc/2024/Conference — Submitted to ICLR 2024_

### Official Review · Reviewer_Up8V · 2023-10-29

**Soundness:** 3 good
**Presentation:** 4 excellent
**Contribution:** 2 fair
**Rating:** 6
**Confidence:** 3

**Summary:**

Due to the depreciation of identifiers causing interference, it's hard to separate groups into treatment and control groups in randomized control experiments. The authors propose VAE-based treatment effect estimators with interference to address the issue. The authors test their model on synthetic data created with the Erdos-Renyi model and the AIRBNB simulations. They conduct extensive experiments comparing different model parameters and compare their work with other methods like HT and DM.

**Strengths:**

- I love the idea and write-up.

  - The authors address an actual problem likely to be faced in randomized online experiments.
  - The authors do a great job of presenting the problem and proposed solution. The introduction is well written, and before they introduce their method, the authors clearly explain the shortcomings of  HT and SUTVA methods. Additionally, the write-up is generally consistent and concise.

- The authors address (and perform experiments) for several different settings, for example, the populations size and impact of neighborhood accuracy on GATES estimation, which was very useful. The graphs are intuitive and comparison with other methods are clear.

**Weaknesses:**

While I like most of the paper, I found a few shortcomings and unclear parts I would like the authors to address;

- Although the authors address the potential issues that could arise from inaccurate neighborhoods, it's mainly focused on the removal/addition of edges. I am curious about a setting where the edge exists but is weak. For example, assume a user owns several devices (e.g., same login credentials for Netflix), however, in principle, there are several *different* users using the same credentials. Would it be better to use a probabilistic matrix and rely on a threshold to decide the neighborhood?


- While the authors address the issue of varied treatments on devices in the neighborhood, I am curious about a case where the actions determining Y are completely different on each device in N (e.g., in the case of shared Netflix passwords accessed by different members.).  How would that affect Y?


- In practical settings, even though users might own different devices, in most cases, they are logged/active on one at a time or restricted to a single access at a time. Would adding the temporal (time) aspect of the model improve treatment estimates?


- There are some write-up issues and typos. Although this can be implied, some variables are not immediately defined when used, e.g., in A2. On page 8, the last sentence before section 5.2 seems incomplete. On page 6, it should |M(i)| >> |N(i)| not |M(i)| >> N(i), page 4 last paragraph, "treatment group in small" should  "treatment group is small", e.t.c.


- Some of the crucial sections are missing. I think having a conclusion/discussion and limitations would help address potential issues that might arise, for example, from unsatisfied assumptions and broader recommendations. I understand this might be a space issue, but if authors can find a way to add them, that might be helpful.


- Other issues, mostlyminor: Authors could improve the font of the figures. What do the authors mean by type in this sentence "if the listing and person have the same type" on page 8? In the age of GDPR and similar privacy measures, device linking might not just fall short but might be an altogether infeasible approach. Lastly, I find the explanation given for the strong assumption (A7) a bit unrealistic.

**Questions:**

I generally like the authors' work and presentation. I have a couple of questions in the weakness section that I would like the authors to address.

---

> ### Author Response · Authors · 2023-11-20
> **Response**
>
> Thank you for your thoughtful review and positive feedback on our paper. We are pleased to hear that you appreciate the idea, write-up, and the overall presentation of our proposed solution. We take this chance to address the shortcomings and questions you have raised.
>
>
> Would it be better to use a probabilistic matrix and rely on a threshold to decide the neighborhood?
> * We draw the reviewers attention to Remark 1 in the paper where we mentioned something along this line of using confidence in an edge to determine neighbourhoods. Conservatively choosing a threshold for inclusion of an edge, will likely result in including a number of extraneous edges, which motivated assumption A7.
> The problem comes from missing an influence which can lead to biased estimates. From Figure 5.3 we can see that there is a potential bias-variance tradeoff with including few edges increasing bias while adding more edges adding variance. One can try to choose the threshold for including an edge based on the importance of these two factors.
>
>
> the actions determining Y are completely different on each device in N ?
> * Assumption A2,A3 support heterogeneity i.e. behaviour on the nodes belonging to the same user need not be similar. Specifically, even for the same treatment allocation z and the same neighbourhoods, two devices can have different true exposures (as $\phi$ depends on $X_i$). However, since our method does not compute true exposures, it cannot determine which nodes are actually influencing the outcomes.
>
>
> Would adding the temporal (time) aspect of the model improve treatment estimates?
> * This is an interesting question, which is a potential future direction of research. While time-series will provide more information, lack of independent errors mean that the result in this work does not ensure identification of the model.
>
>
> Other issues
> * We have added information about this in the Experimental Section, with greater details added in the Appendix. We would also like to draw your attention to the fact that the purpose of this paper is precisely to estimate treatment effects without resorting to identity linking. The superset assumption is motivated exactly by the failure to do identity linking. Since, coarse information can be used under certain contexts, one can create these supersets with such coarse information. Our method maintains privacy, while allowing for estimation of GATE.
>
>
> Typos and Conclusion
> * We have fixed the typo issues and slightly reformatted the paper to address some other points raised by reviewers (such as adding conclusion)

---

> > ### Comment · Reviewer_Up8V · 2023-11-21
> >
> > I would like to thank the authors for the responses and corresponding paper edits addressing the questions raised.

---

### Official Review · Reviewer_ZXoD · 2023-10-29

**Soundness:** 2 fair
**Presentation:** 3 good
**Contribution:** 2 fair
**Rating:** 5
**Confidence:** 1

**Summary:**

The authors provide a method to estimate the global treatment effects. The theoretical analysis is provided and the experiment results verify the correctness of the theoretical results and the effectiveness of the proposed methods.

**Strengths:**

1. The presentation of the paper is good, making it clear for those that are not familiar with the field.
2.  the assumption is shown clearly together with the theoretical analysis.

**Weaknesses:**

1. the contribution of the work should be listed more clearly, compared with the existing methods. The table can be showed if needed.
2. the novel of the theoretical results should be verified. I am not sure which thm is essential and novel in the work. I am not sure whether the prop. is directly from given related works.
3. more experiments results need to be provided with larger dataset and the more complex real world settings


I am not an expert in causal inference therefore some of my questions might not be reasonable. I will change my score according to the following rebuttal and other reviewers.

**Questions:**

1. The analysis in the work is all about the linear model, how about non-linear ones?
2. how are the proposed methods different from the classical VAE, if so, if more advanced ones can be adopted?
3. will the noise distribution influences the methods; for the time series data, will the noise correlation or process take effects?

---

> ### Author Response · Authors · 2023-11-20
> **Response**
>
> We thank you for your review and feedback on our paper. We address your questions and concerns below.
>
> Dataset and Experiment:
> * Since the use case is motivated by application in online experimentation, data for these settings is usually private or unavailable. As such we have to rely on simulations and semi-synthetic data. Furthermore, unlike standard problems like regression, the underlying ground truth is unavailable, necessitating use of simulated data to be able to verify the predictions. This is also the approach used in related works [5]. We would also like to draw your attention to the AirbNb simulation has been generally used for a variety of papers dealing with experimentation on digital marketplace [2,3].
> However, to incorporate the reviewers suggestion, we have evaluated our method on a real observational air-pollution dataset used in [1]. Its setting is slightly different from ours, as it focuses on spatial interference from neighbouring power plants. But we can adapt our method to the use case. We have added these results in the Sec 5.4, and added details in the supplementary material.
>
>
>
>
> Contribution:
> *  The novelty of this work is in demonstrating identifiability of GATE under identity fragmentation which has not been done before. The second novelty comes in considering the case of imperfect knowledge of the device graph which has also not been done. We do so via transforming this problem into a noisy regression problem. The proposition is a novel claim of our work. Its derivation relies on a theorem (stated in Theorem1) from earlier work [4].
>
>
> Questions:
> * Q1) Our model is linear in computation of exposure but non-linear in exposure itself. We draw your attention to the function $g$ in A2 which is a potentially non-linear function applied to the exposure values.
>
> * Q2) Our contribution is in highlighting how identity fragmentation invalidates standard estimators used in A/B testing, and how one can still identify the treatment effects by suitable transformation of the problem. This is important as without identifiability, learning model parameters need not be valid in deriving treatment effects. Next we used the VAE-based method to estimate the treatment effect. In principle other methods for variational inference like normalizing flows etc can also be used for learning the model parameters.
>
> * Q3) The noise distribution as long as independent should be tractable in our setting. While in principle one can posit a model for time-series outcomes as well, it is not clear that identifiability of the mean function holds with non-independent errors. Analysing that is an important direction of future research.
>
>
> References
>
>
> [1] Adjusting for unmeasured spatial confounding with distance adjusted propensity score matching, Papadogeorgou et al. Biostatistic, 2019
>
> [2] Cluster Randomized Designs for One-Sided Bipartite Experiments, Brennan et al. Neurips 2022
>
> [3] Interference, bias, and variance in two-sided marketplace experimentation: Guidance for platforms, Li et al, WWW 2022
>
> [4] Advances in the measurement error literature, Susanne M Schennach. Annual Review of Economics, 2016
>
> [5]  Graph agnostic estimators with staggered rollout designs under network interference. Cortez et al, Neurips 2022

---

### Official Review · Reviewer_QWuy · 2023-10-31

**Soundness:** 3 good
**Presentation:** 1 poor
**Contribution:** 2 fair
**Rating:** 3
**Confidence:** 4

**Summary:**

This paper proposes a new approach for estimating the global average treatment effect (GATE) considering identity fragmentation. The authors conducted both theoretical analysis and experiments to validate their approach, demonstrating its effectiveness compared to standard estimators.

**Strengths:**

S1. An approach to estimate causal effects in the presence of identity fragmentation, enhancing the accuracy and reliability of A/B testing in online platforms.

S2. The approach proves effective in the experiments.

**Weaknesses:**

W1. Poor presentation.

W1-a. Some variables, such as X, lack clear definition or explanation, particularly in their role in estimating true neighbors within the model. This lack of detail or oversimplification can lead to confusion when attempting to understand the motivation and core concepts presented in the paper, significantly impacting its quality.

W1-b. The utilization of the trained model, specifically in the format of equation A2, within the estimation process of the GATE value is not elucidated. This omission introduces ambiguity when attempting to compare it with previous methods in the experimental section.

W2. Unconvincing experiments.

W2-a. The settings appear to intentionally align with the assumptions of the proposed model without practical justification.

W2-b. The experimental environments are not clearly outlined, including specific settings of the baseline estimators and the methodology used for calculating metrics in the figures.

**Questions:**

1 In equation A2, c0, c1, g, and w are referred to as neural network functions, yet no further details are provided. This omission makes it challenging to infer the specific characteristics or behavior of the proposed approach.

2 Regarding your simulation environment:
a. The relationship between the covariate X and the randomly generated "random device graphs" remains unclear. Given the methodology described, it appears that X plays a vital role in determining neighboring relationships. A more detailed explanation is needed to elucidate this connection.
b. The similarity between the equation for generating ground truth and model A2 raises questions. Is there a specific reason for this resemblance, and could it potentially confer an advantage to the proposed model in certain settings? Further clarification on this matter would be beneficial.
c. The computation of the bias metric in the experiment figures is not discernible from the paper alone. Providing insight into how this metric is calculated would enhance the reader's understanding of the experimental methodology.

---

> ### Author Response · Authors · 2023-11-20
> **Response**
>
> We thank the reviewers for their valuable feedback. We have revised the draft to address your concerns, and give summary answers below.
>
> W1-a
> * The novelty of this work is in demonstrating identifiability of GATE under identity fragmentation which has not been done before. The second novelty comes in considering the case of imperfect knowledge of the device graph which has also not been done. We are agnostic to how neighbourhood $M(i)$ is obtained. As such in the experiments for 5.1 we randomly added extra edges in the true-graph to obtain the neighbourhoods. This was hinted in footnote 4 but we see that we can make this clearer. We have revised the draft to make this explicit.
>
> * Regarding the relationship between the covariate X and the randomly generated "random device graphs," X plays no role in determining neighbourhoods $M(i)$. We see that it might not have been sufficiently clear in the manuscript, but our approach is agnostic to how $M(i)$ is obtained.
> In our experiments,X pertains to device-level covariates influencing outcomes conditional on treatments, such as location, phone type, demographic information, or other relevant details available to marketers at the device level. Although X could theoretically be used to determine neighborhoods M(i), our approach remains agnostic to the method of obtaining M, treating it as a given parameter. We have explained this better on Page4 of the revised version.
>
>
> W1-b
> * $\tau$ is defined via Equation 1. One the variational model p_\theta is estimated, $Y_i$ is directly specified by p_\theta(Y_i|X_i,z). We draw your attention to Sec 4.2, where the exact probabilistic model is mentioned. Since p_theta directly specifies the means $\mu_Y$,
> One then estimates $\tau $ as  $1/n \sum \mu(Y|\vec{1},X_i) - \mu(Y|\vec{0},X_i)$, which is just equation 1, where the potential outcomes are replaced by the estimate $\mu$ obtained from p_\theta.
> We have now added these details in Sec 4.2 and provided more information in the supplementary material.
>
>
> W-2 a
> * The experiments in Sec 5.1, are to demonstrate the correctness of our method when all its assumptions hold. The similarity between the generative model in 5.1 and A2, is so that the probabilistic model is correctly specified. We would also  like to highlight that A2 and A3, are general enough to cover many sort of exposure models used in literature [1,2,3]. We have made this clear on Page 5.
> Furthermore, in Section 5.2, we experiment with the airbnb model which does not satisfy A2. (In fact it does not satisfy any exposure assumption). In such a case one can get biased estimates, but as can be seen from Figure 4, all other estimates (except HT) are also biased, and HT requires knowledge of the true graph.
>
>
>
>
> W-2 b
> *  DM and Poly-regression in our experiments stems from their common usage as estimators in the literature. Poly-regression fits a polynomial between exposure (typically the number or fraction of treated neighbors) and observed outcomes, while DM represents the classic difference in mean estimate discussed in Section 3 under SUTVA. We have incorporated this information into the supplementary material, providing additional experimental details for clarity.
>
>
>
>
> Questions:
>
>   1 We used single layer MLPs for all the functions. We have added this in the revision (Remark 3)
>
>
>   2a X plays no role in device graph generation. We independently create a device graph and populate covariates X at the nodes. While in principle one can use the node level X to guess the device graph, we simply assume the existence of the device graph. This is because, as long as A1-7 hold, the model remains identifiable, irrespective of how the estimated device graph $M$ was created. That said practically, having poorer/dense graphs adds significant variance to the estimates. Since we assume $M$ to be given in all our experiments, covariates X are simply used as additional covariates in specifying the mean function parameterizing the outcomes.
>
>   2b Bias is computed as the estimated value $\hat{\tau} - \tau$, where $\tau$ is the true treatment effect. In both experiments, we have access to the underlying generative model, which is used to produce data.As such the true treatment effect can be directly obtained. In the figures we plot the relative bias i.e. $(\hat{\tau} - \tau)/\tau$. Relative RMSE is computed in a similar way as root mean square error between $\hat{\tau}$ and $\tau$, and then normalized by $\tau$
>
>
> References
>
> [1] Estimation of causal peer influence effects, Toulis and Kao, ICML 2013
>
> [2] Cluster Randomized Designs for One-Sided Bipartite Experiments, Brennan et al. Neurips 2022
>
> [3]  Graph cluster randomization: Network exposure to multiple universes, Ugander et al, KDD 2013
>
> [4] Graph agnostic estimators with staggered rollout designs under network interference. Cortez et al, Neurips 2022

---

> > ### Comment · Reviewer_QWuy · 2023-11-23
> > **Thanks for your reply**
> >
> > The reviewer thanks the authors for their response. My concerns in W2 remain, so I will keep my original score.

---

### Official Review · Reviewer_iRkW · 2023-11-06

**Soundness:** 3 good
**Presentation:** 2 fair
**Contribution:** 2 fair
**Rating:** 5
**Confidence:** 4

**Summary:**

This paper proposes a mechanism to perform A/B testing and estimating the global average treatment effect (GATE) in a setting where users interact with online service via multiple devices and the precise mapping of users to devices is unknown, i.e., there is identity fragmentation. The methodology rests on the key assumption that, for each user, a superset of their real devices is known. The paper ventures to show that GATE is possible under such a setting and proposes a good estimator.

**Strengths:**

The paper makes an interesting contribution in an active research area.
Eventually, the results present the developed method to show less bias than others.

**Weaknesses:**

The core assumption is the availability of a model M which provides information on the true underlying user-device graph adjacency matrix A, in the sense that one can get the predicted or assumed neighbors of a device. These neighbors are assumed to be always superset of the true neighbors, as stated in Equation A7, in conformity with the assumptions stated elsewhere in the paper. This assumption is justified by a geographi argument. However, this assumption is not revisited again in Section 5, where experiments are presented. It is not clear what extent those supersets are meant to have. Perhaps that related to the strength of interference r, yet that is not clearly stated. In Section 5.3 the size of the fraction of extraneous neighbors in M(i) is eventually taken in consideration. However, this parameter was not discussed in previous experiments. The notion of extraneous neighbro is not discussed prior to that.

Parts of the paper are incomplete. Section 5.1 ends abruptly. Conclsusions do not exist.

**Questions:**

Why is M(i) not discussed in the first experiments?

---

> ### Author Response · Authors · 2023-11-20
> **Response**
>
> We thank the reviewer for giving us valuable feedback. We answer your questions and concerns below.
>
> Regarding M(i) in Section 5.1
>
>  * a) We had trimmed some content out to meet the page-limit requirements. In Footnote 4 in Section 5.1, we mention that there were up to 100 extraneous nodes, but we see that the exact details could be made clearer. For each node i in the graph, we randomly select between 10 and 100 additional non-neighboring nodes to include in M(i). We have edited Section 5.1 and gurther details have been incorporated into the Appendix. Since we did not explicitly control the size of $M$, experiments in 5.1 do not incorporate the how size of $M$ effects the estimate, and are to demonstrate the validity of our approach across different model parameters when all the assumptions A1-6 are satisfied.
>   * b)  While the size of $M(i)$ has no direct relation to the strength of interference $r$, you have correctly noted that the extent of those supersets might depend on the strength of interference. The dependence of model behaviour on their interplay is not analyzed in Section 5, because, all  the baseline interference aware estimators need the exact neighbourhood information, which precludes analyzing the varying coverage of $M$ . As such in those experiments we have not considered analyzing behaviour with $M$. The interplay between r and superset size is instead considered in Section 5.3. Figure 5a, shows estimation by our model with different values of r. One can see that lower $r$ can support larger (and hence less accurate $M$). We have edited the paper draft to make this clearer, and provided greater details in the supplementary material.
> *  c) $r$ quantifies the ratio of the effect of the ego-node (or the node itself) vs the impact of other nodes in the true neighbourhood $N_i$ on the outcome. We have also added details of what $r$ is on Page 8 (Sec 5.1).
>
> Regarding Conclusions:
>  *   We have fixed the ending of Section 5, and at the reviewers request we have also added a concluding section.

---

### Author Response · Authors · 2023-11-22

We thank all the reviewers for their valuable time to provide constructive feedback regarding all aspects of the draft. We were glad to know that the reviewers found the problem interesting (#iRkW, #Up8V), well presented (#ZXoD, #Up8V) and important  (#Up8V) and our  theoretical analysis intuitive and useful (#ZXoD, #Up8V).

Based on all feedback and suggestions, we have updated the draft. We provide a summary below:

	- Adding additional experiment on a real world observational data
	- Adding description of baseline estimators
	- Specifying exact contributions
	- Mentioning limitation and concluding remarks
	- More experimental details
		○ Description of simulation and datasets
		○ Details of networks used
	- Updated reference style
	- Other minor typo/fixes/suggestions.

---

### Meta-Review · Area_Chair_J4Ce · 2023-12-11

**Metareview:**

Thank you for your submission. While the paper is appreciated for its contribution and clarity in certain aspects, it requires a more detailed and clear presentation, especially regarding assumptions, experimental design, and methodology.

To summarize some of the issues identified by the reviewers:
- Presentation issues, including undefined variables and unclear methodologies (Reviewer QWuy).
- The experimental design and results are questioned for their practical relevance and clarity. The settings seem to overfit the model's assumptions, and there is a lack of detail about the experimental environments (Reviewer QWuy).
- The paper's contribution compared to existing methods is not clearly articulated (Reviewer ZXoD).

Addressing the identified weaknesses and questions from the reviewers will improve the paper's quality and relevance.

**Justification For Why Not Higher Score:**

The idea presented in the paper has potential, but the paper does not seem ready to be published given the identified issues.

**Justification For Why Not Lower Score:**

N/A

---

### Decision · Program_Chairs · 2024-01-16

Reject